# WHEN CAN YOU TRUST LARGE LANGUAGE MODELS?

## ABSTRACT

Quantifying neural network model uncertainty is a difficult problem that has far-reaching implications on our ability to improve model reliability. Uncertainty quantification is especially difficult in the context of LLMs and autoregressive models, as standard methods for uncertainty measurement that apply to single outputs often fail to capture the semantic complexity of the entire autoregressive output. To remedy this gap, we introduce TRUST (Temperature-Related Unambiguity via Similarity Tracking) scores, a novel approach for quantifying LLM uncertainty which reasons about uncertainty *across the entire model output* rather than being limited to a small number of subsequent tokens. TRUST scores take advantage of the natural semantic branching of LLM outputs for nonzero temperatures, and calculate uncertainty based on semantic similarity of multiple output rollouts for an LLM model. We show that TRUST outperforms industry standard uncertainty methods within complex multi-token language tasks like predicting math problem difficulty, and also can be distilled into efficient forward-pass models for easy inference. Crucially, TRUST scores can be calculated with nothing more than standard LLM calls and require zero white-box access to model internals.

## 1 INTRODUCTION

One of the most unfortunate limitations of modern AI models is that they do not know what they do not know. This is unfortunate because, especially within the space of large language models (LLMs), model uncertainty can help us analyze the root causes of model errors (Kalai et al., 2025), train models proactively to close performance gaps (Kendall et al., 2018), or set up early alarm systems for model misbehavior (Weiss & Tonella, 2021). Unfortunately, current language models cannot reliably express their own confidence levels (Xiong et al., 2023), and are well-documented as often being outwardly confidently incorrect (Kidd & Birhane, 2023). Especially as LLMs begin to be adopted by safety-critical industries like finance (Easin et al., 2024) and healthcare (Singhal et al., 2025), these types of uncertainty detection methods will become indispensible to ensure AI continues to behave robustly.

As a result, practitioners will often fall towards statistical measures like entropy (Shannon, 1948) and max softmax probability (MSP) (Hendrycks & Gimpel, 2016; Pearce et al., 2021). However, these statistical measures often struggle in the LLM context, because LLMs by and large remain *autoregressive*, and each prediction is just the next token in a potentially long sequence of text. More sophisticated methods like multi-prediction ensembling variance (Malinin et al., 2019; Fadeeva et al., 2023) suffer from the same limitation. Some recent multi-token methods like Semantic Uncertainty (Kuhn et al., 2023) show promising initial abilities to reason about uncertainty on semantic multi-token outputs, but are often impractical, limited in scope due to fixating exclusively on fact retrieval settings, and tested only on accuracy benchmarks as opposed to true uncertainty benchmarks. Given how pervasive LLMs have become in current AI production systems, it has become absolutely critical to provide a practical, general solution to multi-token uncertainty measurement.

Although LLMs are not very proficient at directly expressing uncertainty, they have been shown to be (1) adept at semantic comparisons between different pieces of text (Chen et al., 2024), and (2) the output text will have natural sampling variance/uncertainty given a nonzero temperature. We thus transform these two observations into a simple but effective algorithm for determining LLM uncertainty: the TRUST (Temperature-Related Unambiguity via Similarity Tracking) score.

TRUST scores are computed by sampling multiple candidate outputs from an input, and then using a separate LLM judge model to compute an average pairwise semantic similarity between the IID

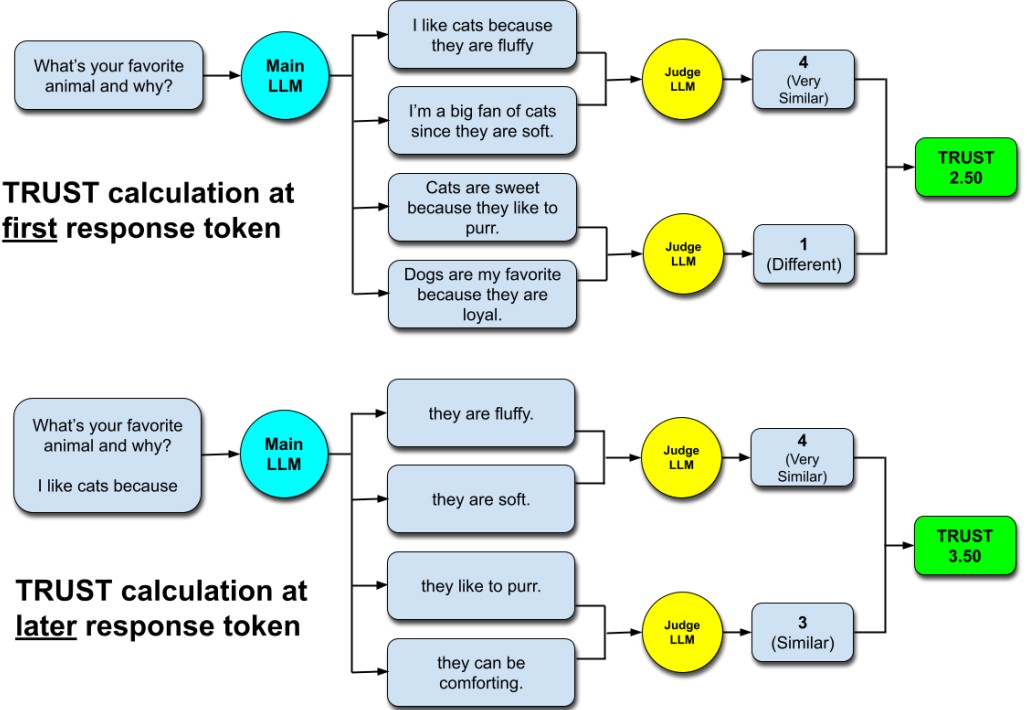

Figure 1: Diagram of calculation of TRUST scores both at the first token and at a later token. A semantic similarity scale from 1 to 5 is assumed, with 5 being most similar. Semantic similarity is judged on the entire response, even if the response was partially locked at input (as in the bottom example, which forces all responses to always be about cats). This leads to generally higher scores as more of the response tokens are locked.

sampled responses. In this way, TRUST scores are sensitive *only* to the semantics of the text output and is agnostic to any architectural details of the generating models. Crucially, calculating TRUST scores does not require any access to the internal state of the model (e.g. logits, activations). Unlike conventional methods such as entropy or softmax, we can therefore use TRUST scores regardless of token sampling strategy.

We will demonstrate that TRUST scores are superior to standard measures of uncertainty in simple settings, and will greatly outperform within more semantically complex settings. In the single-prediction limit, TRUST scores are related to max softmax probabilities (MSPs), but in multi-token settings TRUST scores outperform MSP by a wide margin. TRUST brings uncertainty estimation into the modern LLM era by leveraging the strong semantic understanding abilities of foundation models.

Our main contributions within this work are as follows:

1. We introduce TRUST scores, a metric to estimate LLM uncertainty that can operate in black-box settings and takes into account the semantics of the entire LLM output.

2. In certain limits, we derive a simple relationship between TRUST and maximum softmax probability scores.

3. We show that TRUST scores are superior to standard methods in simple settings and strongly outperform baselines in more complex semantic settings.

4. We show TRUST scores can be used as a training target and distilled into an efficient $\widehat{\text{TRUST}}$ model. The distilled model is competitive with raw TRUST scores and is much faster to compute.

## 2 RELATED WORK

**Uncertainty Estimation in LLMs** Uncertainty estimation has been a long-standing research problem within the AI literature, due to neural networks being poorly calibrated and often confidently incorrect (Guo et al., 2017). Most commonly, LLM uncertainty still uses standard white-box statistical methods like entropy (Wang et al., 2022), max softmax (MSP) (Liu et al., 2023), or variations thereof (Kuhn et al., 2023). Uncertainty detection in LLMs is also often packaged with hallucination detection (e.g. (Manakul et al., 2023; Rawte et al., 2023)), but these problems are not explicitly causally related: a model being uncertain may not lead to incorrect outputs, and vice versa. There have also been attempts to have LLMs directly express their uncertainty (Lin et al., 2022), but these methods tend to lead to false signals and overconfidence (Xiong et al., 2023). However, one advantage of the direct expression approaches is that they do not require white-box access to model internals; TRUST scores operate in black-box mode while enjoying a level of mathematical groundedness usually associated with the white-box statistical methods.

**Uncertainty for Long-Tail Detection** Long-tail detection is intimately tied to uncertainty, as the long-tail of the distribution is generally attributed to high epistemic uncertainty. Uncertainty signals have been used to detect the long-tail in the gradient channel (Chen et al., 2022), using ensembles (Lakshminarayanan et al., 2017; Vyas et al., 2018), through direct multitask prediction (Li et al., 2022), or using scalar uncertainty-related statistics like maximum softmax probability (Hendrycks & Gimpel, 2016). Uncertainty measures can also be used for adversarial example detection (Smith & Gal, 2018). However, we note that these methods generally restrict the problem to uncertainty of a single prediction, which is not the complete picture in an autoregressive sequence generation setting.

**Uncertainty for Model Training and Inference** Uncertainty can also be used to actively benefit the training and deployment phases for deep models. Active learning often uses uncertainty signals (Yang & Loog, 2016; Shi et al., 2020) or loss value prediction (another form of uncertainty) to inform training dynamics (Yoo & Kweon, 2019). Uncertainty can also be used directly in multitask learning either as a direct loss multiplier (Lin et al., 2017; Kendall et al., 2018) or within gradient space (Chen et al., 2020) to produce more generalizable models. Uncertainty can also be used in continual learning approaches to automatically improve model quality over time (Ahn et al., 2019; Jha et al., 2023). We note that there is still a relative scarcity of work in the LLM space using uncertainty to directly benefit training; we hope that TRUST scores will help fill this gap by providing a much more semantically complete view of autoregressive uncertainty.

## 3 METHODS

### 3.1 UNCERTAINTY FOR COMPLEX OUTPUTS

Generally speaking, uncertainty is a mapping between a model output $\mathcal{M}(x) = \hat{y}$ and a scalar $U$. In the language of LLMs, the mapping is often between a prefix and the next token, and standard methods like entropy (Shannon, 1948) and max softmax probability (MSP) (Hendrycks & Gimpel, 2016; Pearce et al., 2021) are often limited in application to these next tokens.

In natural language settings, such restrictions to next-token prediction are clearly insufficient, as semantic meaning within natural language is encoded only in the full output. Due to the autoregressive nature of most language generation models, white-box uncertainty metrics like entropy and MSP are difficult to extend in a simple way to multi-token semantic meanings, due to the intractability of the number of possible token rollouts.

One notable attempt to extend entropy to multi-token semantic outputs is Semantic Uncertainty (Kuhn et al., 2023), which has shown to be effective at predicting model accuracy within a variety of fact retrieval settings like TriviaQA (Joshi et al., 2017) and CoQA (Reddy et al., 2019). However, a few issues within the design of Semantic Uncertainty make it unsuitable for general uncertainty prediction for LLMs. From a practicality perspective, the method requires multiple entailment calculations from a custom entailment model to group potential outputs into equivalence classes. But more importantly, its focus on concrete sentences as the main input unit for uncertainty measurement does not generalize well beyond output text that is decomposable into discrete factoids, as we will demonstrate empirically later.

In general, a fundamental weakness of many prior uncertainty works like (Kuhn et al., 2023) is that they focus on correlations between uncertainty and *accuracy*. We presume that part of this design decision lies in practicality, as accuracy benchmarks are much more abundant than true uncertainty benchmarks. But we feel it is crucial to decouple accuracy from confidence, and to focus on the latter in treatments of uncertainty estimation. We will show that on the more difficult benchmarks focused on uncertainty, prior methods like Semantic Uncertainty tend to underperform.

TRUST is designed to be general and applicable to complex long outputs by moving away from factoid decomposition and detecting uncertainty at the token level (while still taking multi-token semantics into account). It is designed to be practical as it only requires off-the-shelf LLMs to calculate and can also be distilled down into lightweight language models like BERT. Moreover, it is tested in proper uncertainty settings rather than focusing on accuracy measurements, which demonstrates that TRUST is calibrated as a good measure of actual model confidence versus being a rough proxy for model prediction accuracy.

### 3.2 Computing TRUST Scores

We proceed with a few key observations on uncertainty estimation in language settings:

1. Output variance prediction for language models must be computed at the semantic level, rather than the token ID level. This means that any valid method must be insensitive to semantically-invariant dimensions such as synonyms or word choice.

2. LLMs are generally ineffective at directly expressing their own uncertainties (Xiong et al., 2023), but are proficient at binary semantic comparison (Chen et al., 2024).

3. The semantic structure of general language outputs is complex, with token-level being the only consistent decomposition of language across models.

4. Language models have natural sampling variance at nonzero temperature.

One major advantage of autoregressive generation settings in uncertainty estimation is that there is natural sampling variance within generation in the form of the temperature scaling parameter $\tau$. Thanks to temperature scaling, we do not need to rely on ensembling to generate multiple potential model outputs (which itself is impacted by training hyperparameters and other extraneous factors), but can directly generate multiple model outputs from the same model at nonzero temperature.

We take advantage of the temperature-induced output variance of a model $\mathcal{M}$. Denote the input query of the model as $\mathbf{q}^{(j)}$ and a partial completion of $t$ tokens ($t$ might be 0) as $\mathbf{r}^{(j,t)}$. The complete prefix input into a language model is the concatenation $\mathbf{x}^{(j)} = \text{Concat}(\mathbf{q}^{(j)}, \mathbf{r}^{(j,t)})$. We then generate $2N$ completions at temperature $\tau > 0$, usually until a stop token is encountered. Denote these completions $\mathbf{y}_i^{(j)}$, $i \in (1, \ldots, 2N)$. We then initialize a judge model $J$, usually a pre-trained LLM, which produces semantic similarity scores when comparing two pieces of text. The TRUST score at token index $t$ for example $j$ is defined as

$$\text{TRUST}_{j,t} = E_{i=1,\ldots,N} \left[ J \left( \text{Concat}(\mathbf{r}^{(j,t)}, \mathbf{y}_{2i-1}^{(j)}), \text{Concat}(\mathbf{r}^{(j,t)}, \mathbf{y}_{2i}^{(j)}) \right) \right] \tag{1}$$

The TRUST computation is schematically illustrated in Figure 1. To save compute, generally we will only generate $N + 1$ completions and make pairwise comparisons between $i$ and $i + 1$ in the above expectation rather than between $2i - 1$ and $2i$.

In simpler terms, we take the $2N$ completions and prepend them with a partial candidate completion up to the desired token to form the full response (everything after the input query). We then pairwise compare these responses and the TRUST score is the average across these pairwise comparisons. The resultant score is attached to that specific token position ($t$). We note that in this formulation, the TRUST score is an average of similarity scores, and thus the higher it is the lower the model uncertainty is at that token position.

We position TRUST scores as a measure of total uncertainty, without any explicit decomposition into epistemic or aleatoric uncertainty. Further discussion of this point can be found in Section 5.

### 3.3 TRUST MODELING

Although Section 3.2 allows us to generate TRUST scores, generating $N$ trial completions at each token position to produce TRUST scores can be expensive and time-consuming. Thus, we also test performance of a model trained on TRUST scores. These can be simple language predictive models, and we will use a BERT (Devlin et al., 2019) model within this work. We will show that a predictive model trained on TRUST scores is competitive with using the raw TRUST scores themselves, demonstrating that we can avoid the computational downsides of having to compute TRUST scores at inference time.

### 3.4 THEORY

We can show that TRUST scores for single-token prediction and under mild conditions are related to the squared maximum softmax probability (MSP) (Hendrycks & Gimpel, 2016), a measure of uncertainty that has been widely used in industry since its invention.

**Theorem 1.** *Take a sequence prediction model $\mathcal{M}$ which autoregressively predicts the next sequence element out of $V$ possible elements $y$ through sampling of some probability distribution with temperature $\tau$ $p(y; \tau)$. Denote the second highest probability as $p_2(y; \tau)$. Assume we only predict one next element in the sequence and have an ideal judge model $J'$ that produces $J'(y_i, y_j) = 1$ if $i = j$ and $J'(y_i, y_j) = 0$ if $i \neq j$. If TRUST is formed through a single pairwise comparison between IID sampled next sequence elements, then*

$$E[\text{TRUST}] = (\max_i(p_i(y; \tau)))^2 + O(p_2^2(y; \tau)) \geq (\max(p(y; \tau)))^2 \tag{2}$$

*Proof.* Given a single next-element prediction, the probability that the two elements $y_i, y_j$ sampled are identical is

$$p(y_i = y_j) = \sum_{i=1}^{V} p(y_i)^2 \tag{3}$$

Because our judge model is ideal, this is also the probability that the judge will return a score of 1. Thus, we have

$$E[\text{TRUST}] = p(J'(y_i, y_j) = 1) \tag{4}$$

$$= \sum_{i=1}^{V} p(y_i)^2 = (\max_i(p_i(y; \tau)))^2 + O(p_2^2(y; \tau)) \geq (\max_i(p_i(y; \tau)))^2 \tag{5}$$

Thereby proving our result. $\square$

Evidently, TRUST scores applied only to next-token prediction are generally the square of the MSP score in addition to higher order terms, but are always lower bounded by the square MSP. In situations where the MSP is high (which is common), the TRUST score is a close approximation of the square MSP score. Even low MSP is often caused by synonyms or other semantically close tokens, and Theorem 1 would still hold with the slight modification that synonym tokens are clustered and their sampling probabilities considered jointly.

We emphasize that the theory presented here only applies to single-token prediction, and TRUST scores are even more powerful in multi-token settings (which we will demonstrate empirically later). We only provide this theory to show that TRUST scores are well-motivated by strong industry-standard uncertainty baselines.

## 4 EXPERIMENTS

### 4.1 BASELINE METHODS

Our baseline measurements for uncertainty consist of Shannon entropy ($-\sum p_i \log(p_i)$) and Maximum Softmax Probability (MSP). We also compare against ensembling uncertainty in Section 4.3. For more implementation details on baselines, please refer to Appendix A.6.

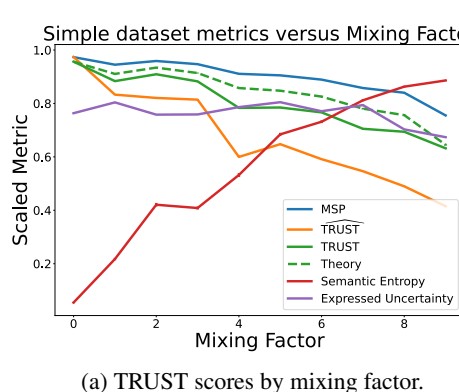
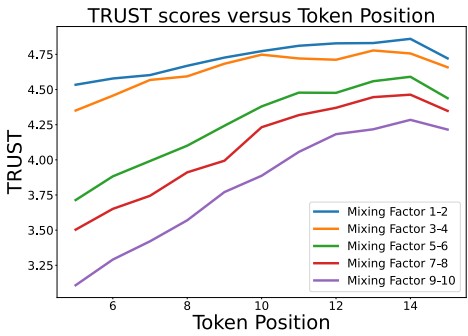

(a) TRUST scores by mixing factor.       (b) TRUST scores vs token position.

Figure 2: (a) MSP and TRUST values vs mixing factor, along with a theory line at MSP$^2$. All metrics are scaled to lie on a [0,1] scale. (b) Average raw TRUST scores plotted against token position. We consolidate data in every other level to make the visualization smoother. As expected, TRUST scores tend to be monotonically increasing as fixing more tokens in the response leads to fewer opportunities for response branching.

Our proposed uncertainty measure will be labeled TRUST for the raw score and $\widehat{\text{TRUST}}$ for the model trained on our TRUST raw scores, as described in section 3.3. In general, the distilled model $\widehat{\text{TRUST}}$ will prove to be competitive with the raw scores.

Completions were generated by GPT-4o-mini (Hurst et al., 2024) and Llama-3.1-70b (Dubey et al., 2024). The judge model was always set to GPT-4o to ensure we were using a high-performing model. Our judge scores are on a scale of 1-to-5, with 5 being the most similar. Judge prompts can be found in Appendix A.4. Unless otherwise noted, all TRUST generations were performed at temperature $\tau = 1.0$. We discuss this choice in Appendix A.1.

## 4.2 SIMPLE UNCERTAINTY PREDICTION

For a controlled setting, we generate a synthetic dataset of 80 simple questions and ten distinct responses to each question. These responses are preferences on a variety of topics, such as pets, outdoor activities, and cooking recipes. We include the full dataset as part of the Supplementary Materials, and display here a few sample responses from the dataset:

> **Q: What's your favorite pet?**
> - I like cats because they are playful and independent.
> - I like dogs because they are loyal and playful.
> - I like parrots because they are colorful and can mimic sounds.

We then form ten datasets of mixing factor $M \in (0, \ldots, 9)$. A dataset with mixing ratio $M = i$ is composed of $(100 - 10M)\%$ of the first possible response to each question, with the remaining dataset split uniformly across all other responses. This means increasing $M$ increases the dataset entropy until saturating at the uniformly distributed limit.

Each dataset of different mixing factor is then fine-tuned into a medium-sized LLM using LoRA (Hu et al., 2021) adapters in ten isolated trials over the ten resampled datasets respectively. Specifically, we fine-tune the Llama 3.1 8B model using LoRA adapters with the standard next-token prediction objective. We then take the trained models and test how the various baselines described in Section 4.1 correlate with the mixing factor $M$. Final uncertainty scores for each baseline were calculated as an average of token-level uncertainty scores across a contiguous window of token positions within the response; this window was treated as a hyperparameter and tuned for each baseline.

The results are shown in Table 1. We note that the trained model $\widehat{\text{TRUST}}$ performs nearly as well as the raw TRUST scores and the Semantic Entropy scores, which shows that TRUST scores

| Correlation $|\sigma|$ to Mixing Factor ↑ | | | | | | |
|---|---|---|---|---|---|---|
| $\widehat{\text{TRUST}}$ | TRUST | SE | MSP | MSP$^2$ | Entropy | EU |
| **0.969±0.001** | **0.976±0.001** | **0.977±0.001** | 0.935±0.003 | 0.947±0.003 | 0.956±0.002 | 0.548±0.008 |

Table 1: Correlation coefficients of candidate uncertainty metrics vs Mixing Factor. SE is Semantic Entropy and EU is Expressed Uncertainty. MSP-Entropy is omitted with value $0.900 \pm 0.006$.

can be treated as training targets and distilled into efficient models. We also include MSP$^2$ as a baseline following the discussion in Section 3.4. In this case, all methods perform fairly well in this simple problem setting, but TRUST outperforms all other baselines except for semantic entropy which performs at parity, which we expected as semantic entropy is optimal in settings with simple declarative outputs. We note that even though in certain limits we know TRUST is related to MSP$^2$, even in this simple setting with multi-token outputs we already outperform the single-output methods.

We note that correlations can depend on the functional form of the correlants: correlating $x$ with $y$ will give different results than correlating $x^2$ with $y$. In our case, the comparisons in Table 1 are valid to leading order because our MSP scores are generally close to 1 (See Figure 2(a)) and thus all candidate metrics admit a linear approximation. However, we do include comparisons to MSP$^2$ and MSP-minus-Entropy, because TRUST is quadratic in MSP (Section 3.4) and MSP-Entropy$= MSP + \sum_i p_i \log(p_i) \approx$ MSP+MSP(MSP-1) = MSP$^2$. TRUST still outperforms all transformed versions of these metrics.

We display average metrics in Figure 2, along with a theory line in Figure 2(a) to show that the theory posited in Section 3.4 tracks very closely with our observed TRUST scores. We note that even though average TRUST and the theory line are on top of each other, individual TRUST scores still outperform MSP$^2$ which indicates that the TRUST scores are capturing some additional semantics of the problem that single-prediction methods cannot capture. Additional curves showing other baseline scores like entropy are in Appendix A.2.

### 4.3 DIFFICULTY PREDICTION ON THE MATH DATASET

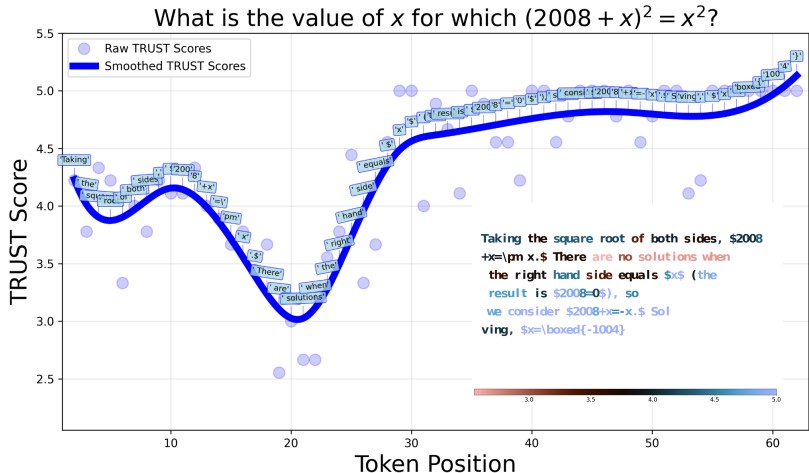

Figure 3: Example visualization of TRUST scores on entries in the MATH (Hendrycks et al., 2021) dataset. Higher similarity equates to lower uncertainty. In the colorized example, tokens that are colored red correspond to lower TRUST scores and higher uncertainty.

We now move to a much more complex setting: predicting the difficulty level of math problems in the MATH dataset (Hendrycks et al., 2021), which consists of open-ended math problems labeled with granular levels of difficulty (1 to 5).

| LLM | $\widehat{\text{TRUST}}$ | TRUST | SE | EU | MSP | Entropy |
|---|---|---|---|---|---|---|
| | | MATH Difficulty Mean Squared Error (MSE) ↓ | | | | |
| GPT | **1.18 ± 0.04** | **1.20 ± 0.04** | 1.45 ± 0.01 | 1.52 ± 0.26 | 1.47 ± 0.04 | 1.46 ± 0.04 |
| Llama | **1.22 ± 0.05** | **1.22 ± 0.02** | 1.44 ± 0.05 | 1.37 ± 0.05 | 1.42 ± 0.03 | 1.58 ± 0.13 |

Table 2: MSE for the difficulty prediction task in the MATH dataset. GPT is gpt-4o-mini and Llama is llama-3.1-70b. SE is Semantic Entropy and EU is Expressed Uncertainty. Ensemble method for Llama is omitted with value 1.43 ± 0.05.

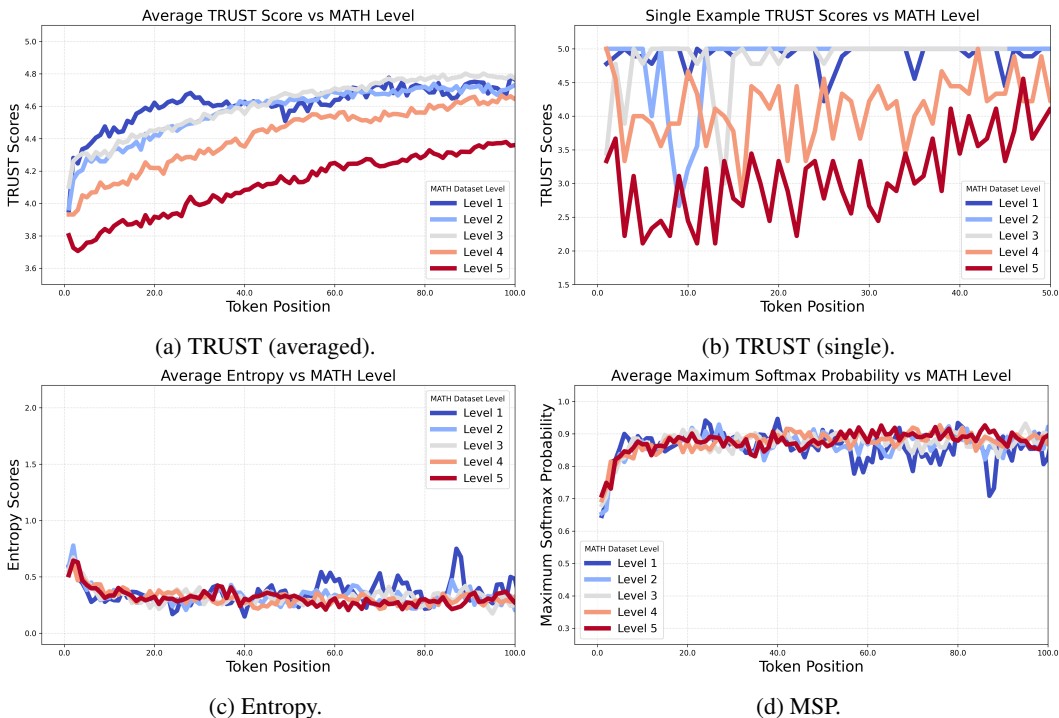

(a) TRUST (averaged).      (b) TRUST (single).

(c) Entropy.      (d) MSP.

Figure 4: Comparing different measures of uncertainty vs token position. (a) The average TRUST curves as well as (b) the noisier single example profiles show clear separation between levels, while (c) Entropy and (d) MSP fail to discriminate between different MATH levels.

For these experiments, we also test against the mean pairwise KL-divergence of a model ensemble, like in Fadeeva et al. (2023). The KL-divergence ensemble metric is only calculated for Llama as we do not have white-box access to GPT-4o for ensemble training.

After computing raw uncertainty scores, we trained a single projection layer to predict the MATH difficulty score from just the raw scores. As in Section 4.2, we curtailed the inputs for each candidate method to only include tokens within a empirically determined contiguous window; for TRUST we found tokens from position 5 to 15 worked well. Results are displayed in Table 2 for sets of experiments where we used GPT-4o-mini and llama3.1-70b for response generation. We observe a significant improvement of performance of TRUST scores in predicting difficulty in the MATH dataset. A visualization of outputs from the MATH dataset is also shown in Figure 3

In Figure 4, we see that even by eye the separability of the different difficulty levels by TRUST scores is apparent, while the signal is washed out or nonexistent for all other baselines. Once again, the BERT model trained to predict raw TRUST scores performs quite well. These results demonstrate that in complex real-world scenarios, TRUST scores cover a fundamental performance gap inherent in standard single-prediction methods.

One interesting observation within the curves of Figure 4 is that separability of different levels for TRUST is most pronounced within the first ≈twenty tokens of the response, with scores tightening

later in the response. This effect is intuitive, as most variability of language model responses are likely concentrated close to the beginning of a response.

However, semantic entropy and expressed uncertainty lag behind significantly in this complex dataset. We hypothesize that semantic entropy struggles with long solution derivations in the MATH dataset because each step in the derivation depends on previous steps, a reality that entailment and factoid decomposition fail to capture.

## 5 DISCUSSION

One common discussion within the AI uncertainty literature, including for LLM uncertainty (Yadkori et al., 2024), is whether proposed methods are more sensitive to epistemic or aleatoric uncertainty. We position TRUST as an estimate of *total uncertainty*, especially as the classic decomposition of uncertainty into its aleatoric and epistemic components can be ill-defined within discrete prediction spaces like language (Wimmer et al., 2023; Smith et al., 2024). We note that the experimental setting explored in Section 4.2 seems like a classic example of aleatoric uncertainty (i.e. *single input $\mapsto$ multiple valid outputs*]. In contrast, the MATH dataset setting in Section 4.3 is a good example of epistemic uncertainty measurement, as the data points within the dataset are all well-defined, solvable math problems that are in-principle learnable by a model. Thus, we have shown that TRUST is sensitive to all sources of uncertainty.

In the future, we will explore further decompositions of TRUST scores into model and text prefix components, following the classic Bayesian decomposition $P(\text{out}|\text{in}) = P(\text{out}, \text{in})/P(\text{in})$. These studies will extend TRUST to capture uncertainties that are invariant to the generating LLM, while in this current work we always generate TRUST with respect to a specific data source and model.

We reiterate that TRUST scores are sensitive to model uncertainty, *without consideration of whether the model is accurate*. Accuracy and uncertainty have often been tested together (Manakul et al., 2023), and loss prediction has been used as a proxy for uncertainty detection (Yoo & Kweon, 2019), but a model's uncertainty is correlated with while not being causally related to model accuracy. We selected settings within both our toy example and MATH experiments to isolate the uncertainty signal away from any accuracy signals, and showed that TRUST scores excelled in those settings.

TRUST scores are a total uncertainty metric tailored for the LLM era that extends well-motivated statistical methods like MSP to multi-token outputs by leveraging the semantic understanding of LLMs, while requiring zero access to model internals. Methods like TRUST represent a "best of both worlds" approach, where LLM semantic analysis augments statistically grounded metrics to produce uncertainty measurements that are both theoretically sound while being effective on the challenging high-dimensional domain of natural language.

## 6 CONCLUSION

In this work, we introduced TRUST scores, a novel method that sets a new state-of-the-art for LLM uncertainty estimation by enabling semantic reasoning across the entire LLM output while not requiring any special access to model internals. We showed that TRUST scores are related to maximum softmax probabilities in single-prediction limits, and are effective probes of total uncertainty in both simple language settings and complex settings like math problem difficulty prediction. Importantly, the superiority of TRUST scores is especially pronounced within more complex settings, which sets TRUST up as a good choice within high-dimensional language tasks. We also show that TRUST scores are learnable by shallow language models, which allows for cost-effective real-time inference.

TRUST scores are a simple, elegant extension of industry-standard uncertainty methods tailored for the era of LLMs, and demonstrate the promise of melding the impressive semantic understanding of modern LLMs with principled statistical ensembling for robust uncertainty estimation. Perhaps most importantly, TRUST scores are **exceedingly simple** to calculate, and any practitioner with access to an LLM endpoint can compute them with very little engineering overhead or any white-box access to model internals. We hope TRUST will make uncertainty estimation much more accessible and accurate for LLM applications, which is especially critical if we aim to keep LLMs safe as they permeate more and more aspects of our digital lives.

## 7 REPRODUCIBILITY STATEMENT

The authors of this work were careful to detail all processes and assumptions and ensure that the results within this manuscript are reproducible. All datasets used are either public or in the case of the experiments within a controlled setting (Section 4.2) included in the supplementary materials. All implementation details necessary to reproduce our work are provided in Section 4 for specific experimental settings, and Section 3 for TRUST score computation. Appendix A.2 and A.4 provides all LLM prompts used in our work, and Appendix A.5 provides all architectural details on how to train a distilled TRÛST model. Details in Appendix A.6 provide exact equations for computations of baselines.

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

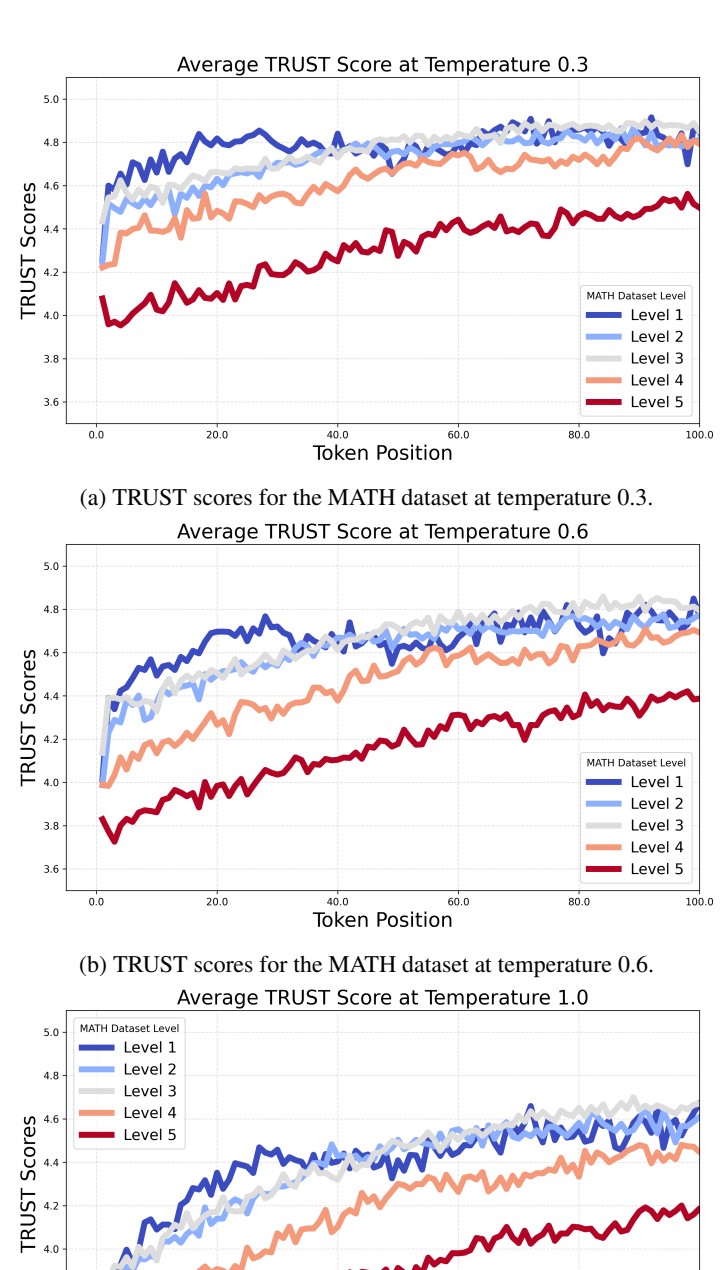

(a) TRUST scores for the MATH dataset at temperature 0.3.

(b) TRUST scores for the MATH dataset at temperature 0.6.

(c) TRUST scores for the MATH dataset at temperature 1.0.

Figure 5: Comparing TRUST Scores across different temperature settings of LLM's.

# A APPENDIX

## A.1 CHOOSING TEMPERATURE

All of our experiments within this work were performed at a set temperature $\tau = 1.0$. We did perform some experiments at other temperatures; for example, in Figure 5 you can see the same experiments on the MATH dataset as in Figure 4 but done at different temperatures of $\tau = 0.3$ and

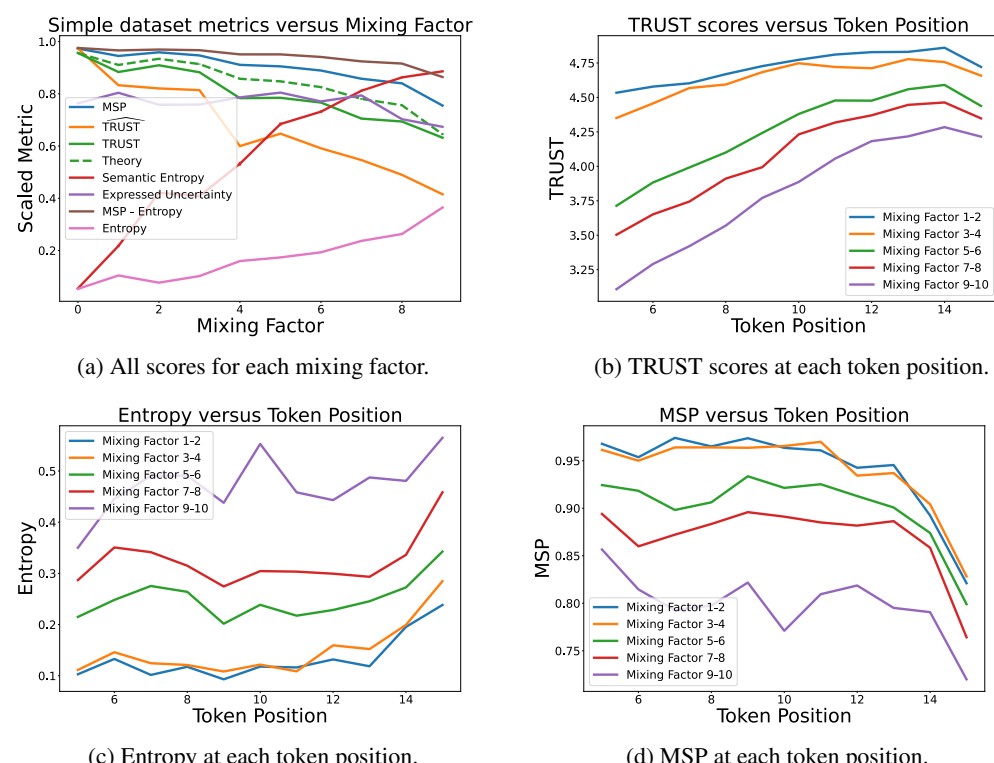

(a) All scores for each mixing factor.

(b) TRUST scores at each token position.

(c) Entropy at each token position.

(d) MSP at each token position.

Figure 6: Comparing different measures of uncertainty across mixing factors and token positions on the simple dataset.

$\tau = 0.6$. In general, the discriminative ability of TRUST seems to be fairly consistent across different temperatures. As such, we picked $\tau = 1.0$ as we knew the temperature will be high enough to induce substantial variability in the response.

In the future, it would be interesting to also see if extensions to TRUST that take information at multiple temperatures into account can perform better. For example, it is clear from Figure 5 that higher temperatures induce lower similarity scores, which is reasonable given that higher temperatures would cause a larger degree of model divergence. Is there a signal hidden within the speed of emergence of this variability as temperature is tuned upwards that we could utilize? These types of second-order temperature effects were out of scope for the current work, where we wanted to exhibit the pure performance of single TRUST scores, but would be interesting avenues for followup research.

## A.2 SIMPLE HUMAN PREFERENCE DATASET - APPENDIX

We show in Figure 6(a) a more complete version of the figure shown in the main paper Section 4.2. The scores shown on the left are raw unscaled scores. We see that at least visibly by eye, both MSP and entropy contain more noise than TRUST , which is consistent with our observation that TRUST tends to overperform on this baseline (Table 1).

The following is the system prompt used with the OpenAI gpt-4o chat completions API when generating questions sequentially to avoid repeating previously generated questions:

> Ask me a short and simple question about my preferences without giving me any options.
> The question should be less than 20 words.
> Do not try to answer the question.
> Below are questions that have been asked before. Please generate a new unrelated question.
> {previous_questions}

The following is the system prompt used with the OpenAI gpt-4o chat completions API when generating answers sequentially to avoid repeating previously generated answers:

> Given the following question and optional reference answers, create a new version of the
> answer with altered preferences that is not similar to the reference answers.
> The answer should be a single complete sentence with simple reasons.
> Make sure to respond with less than {word_count} words.
> Example:
> - Question: What's your favorite pet?
> - Reference answers: I like dogs because they are loyal and playful.
> - New answer: I like cats because they are independent and cuddly.
> Question:
> {question}
> Reference answers:
> {answers}

### A.3 GENERATING TRUST COMPLETIONS WITH THE COMPLETIONS API

The following is the instruction prepended to the question when generating completions in the **MATH dataset**:

> Continue generating the response given the following context, question and partial answer
> such that the total answer is at most {max_words} words.

The following is the instruction prepended to the question when generating completions in the **simple dataset**:

> Continue generating the response given the following question and partial answer such that
> the total answer is at most {max_words} words. The answer should be a single complete
> sentence with simple reasons.

### A.4 COMPUTING TRUST SCORES WITH AN LLM JUDGE

The following is the system prompt used to judge the semantic similarity score between two completions for the **simple dataset** with the OpenAI gpt-4o model:

> **Task:** Rate the semantic similarity between Answer 1 and Answer 2 on a scale of 1-5,
> where:
> - 1 = Not similar meanings
> - 2 = Similar meanings with some potential dissimilarities
> - 3 = Very similar meanings with slight differences
> - 4 = Almost the same meaning with very few differences
> - 5 = Essentially the same meaning, highly semantically similar
> Answer 1:
> {answer_one}
> Answer 2:
> {answer_two}

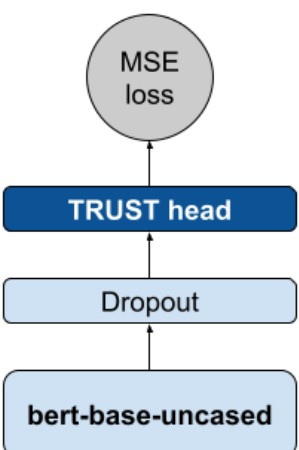

Figure 7: TRUST BERT model architecture with TRUST prediction head

The following is the system prompt used to judge the semantic similarity score between two completions for the **MATH dataset** with the OpenAI gpt-4o model:

**Task:** Rate the semantic similarity between Answer 1 and Answer 2 on a scale of 1-5, where:
- 1 = Completely different meanings, no semantic overlap
- 2 = Mostly different with minimal semantic similarity
- 3 = Some semantic similarity, but notable differences in meaning
- 4 = Very similar meanings with minor differences
- 5 = Essentially the same meaning, highly semantically similar
Answer 1:
{answer_one}
Answer 2:
{answer_two}

## A.5 TRUST BERT MODEL ARCHITECTURE

```
BertTokenSimilarityModel(
  (bert): BertModel(
    (embeddings): BertEmbeddings(
      (word_embeddings): Embedding(30522, 768, padding_idx=0)
      (position_embeddings): Embedding(512, 768)
      (token_type_embeddings): Embedding(2, 768)
      (LayerNorm): LayerNorm((768,), eps=1e-12, elementwise_affine=True)
      (dropout): Dropout(p=0.1, inplace=False)
    )
    (encoder): BertEncoder(
      (layer): ModuleList(
        (0-11): 12 x BertLayer(
          (attention): BertAttention(
            (self): BertSdpaSelfAttention(
              (query): Linear(in_features=768, out_features=768, bias=True)
              (key): Linear(in_features=768, out_features=768, bias=True)
              (value): Linear(in_features=768, out_features=768, bias=True)
              (dropout): Dropout(p=0.1, inplace=False)
            )
            (output): BertSelfOutput(
              (dense): Linear(in_features=768, out_features=768, bias=True)
```

```
              (LayerNorm): LayerNorm((768,), eps=1e-12, elementwise_affine=True)
              (dropout): Dropout(p=0.1, inplace=False)
            )
          )
          (intermediate): BertIntermediate(
            (dense): Linear(in_features=768, out_features=3072, bias=True)
            (intermediate_act_fn): GELUActivation()
          )
          (output): BertOutput(
            (dense): Linear(in_features=3072, out_features=768, bias=True)
            (LayerNorm): LayerNorm((768,), eps=1e-12, elementwise_affine=True)
            (dropout): Dropout(p=0.1, inplace=False)
          )
        )
      )
    )
    (pooler): BertPooler(
      (dense): Linear(in_features=768, out_features=768, bias=True)
      (activation): Tanh()
    )
  )
  (dropout): Dropout(p=0.1, inplace=False)
  (trust_head): Linear(in_features=768, out_features=128, bias=True)
)
```

## A.6 BASELINE METHODS

Here we provide some additional implementation details for all our baseline methods. In terms of calculating some of our white-box statistics like MSP and entropy, conventional LLM APIs can only return a truncated list of log-probabilities containing up to $k$ out of the supported vocabulary $V$. This list is limited to the top 5 log-probabilities from the Fireworks AI API, which we used to query Llama models, or the top 100 log-probabilities from the OpenAI API.

Let $k \in \{5, 100\}$ be the number of log-probabilities returned by these APIs respectively and $y_i$ be the i-th probability in the returned list of top probabilities for any generated token.

After converting the returned log-probabilities to the original softmax values through exponentiation, one can compute the following Partial Entropy for any generated token:

$$PE_k = -\sum_{i=1}^{k} p(y_i; \tau) \cdot log\Big(p(y_i; \tau)\Big) \tag{6}$$

LLMs attribute most of the probability mass to these top k tokens, as evidenced by the high MSP scores we observed throughout all our experiments, so we assume for simplicity with no alternative that the remaining entries that were not returned by LLM APIs follow a uniform distribution. We denote the Residual Probability by:

$$RP_k = 1 - \sum_{i=1}^{k} p(y_i; \tau) \tag{7}$$

Then we divide the residual probability mass among the remaining $(V - k)$ tokens equally to obtain the Residual Entropy:

$$RE_k = -\sum_{i=1}^{V-k} \frac{RP_k}{V-k} \cdot log\left(\frac{RP_k}{V-k}\right) = -RP_k \cdot log\left(\frac{RP_k}{V-k}\right) \tag{8}$$

Lastly, combine the Partial Entropy with the Residual Entropy to obtain the Estimated Entropy:

$$EE_k = PE_k + RE_k \tag{9}$$

Trivially, the Maximum Softmax Probability is represented by:

$$MSP = \max_i p(y_i; \tau) \tag{10}$$

Finally, the Ensemble KL Divergence is the log-adjusted pairwise mean of KL divergences among $h$ number of ensemble prediction head distributions with head indices $a \neq b$:

$$EKL = \binom{h}{2}^{-1} log\left(\sum_{a \neq b}^{\binom{h}{2}} D_{KL}\Big(p(y^a; \tau) \,||\, p(y^b; \tau)\Big)\right) \tag{11}$$

