# OpenReview forum: "When Can you TRUST Large Language Models?"
_ICLR.cc/2026/Conference — Submitted to ICLR 2026_

### Official Review · Reviewer_Bezx · 2025-10-15

**Soundness:** 1
**Presentation:** 2
**Contribution:** 1
**Rating:** 0
**Confidence:** 5

**Summary:**

The authors propose TRUST, a method for measuring the variance of sequences output by an LLM. Focusing on benchmarks that attempt to measure variability in the ground-truth output distribution rather than just correctness, they show that it outperforms a handful of naive baselines. They also show that small, efficient BERT-style models can approximate the TRUST metric.

**Strengths:**

I appreciate that the authors focus on efficiency, which is often overlooked in LLM uncertainty quantification papers, and also emphasize the difference between hallucination detection and uncertainty quantification, which is rarely discussed.

**Weaknesses:**

The glaring issue with this paper is that it does not situate itself properly in the literature. The motivating observations in section 3.2 (namely, that LLM output variance must be computed at the semantic level, that LLMs can't estimate their uncertainty themselves, and that LLMs produce different outputs at nonzero temperatures) are not original to this paper, and indeed are precisely the facts that motivate "semantic uncertainty" (Kuhn et al. 2023, Farquar et al. 2024). That method bears remarkable similarities to this one, but despite having become a universal baseline in this subfield, it is mentioned here only once, and in the related work. Even the idea of training a more efficient model to approximate the metric has been explored already in the semantic uncertainty literature, most notably in Kossen et al. 2024. I think this paper needs to be substantially revised and put in dialogue with this (large) body of work before I can consider accepting it.

Separately, in no particular order:

- The evaluations in the paper are non-standard and, in my opinion, somewhat flawed. The synthetic dataset does not actually elicit meaningful variance in LLM outputs, since each of the possible outputs to each question appears to be completely semantically distinct from the others. Therefore, you're not actually measuring TRUST's robustness along "semantically-invariant dimensions such as synonyms or word choice."
- Fine-tuning a model on the evaluation dataset before evaluating it seems a bit confounding. Fine-tuning (especially on synthetic data) probably collapses the variance of model outputs on prompts like this, which is precisely what we're trying to predict here.
- The idea of using the BERT model to predict TRUST scores outright, rather than just relegating it to the role of predicting similarity scores, seems dubious to me, and the fact that it does well in these evaluations probably says more about the evaluations (which, again, are not standard in the literature) than it does about the BERT model. Some problems that are difficult for one language model are easy for another. If you're using the same BERT TRUST score predictor with no input from the language model being evaluated (this is my interpretation of how BERT was used in the paper), you'll never capture that.
- Section 3.4 is a little strange to me. Why does it matter that TRUST, under ideal conditions, is strictly greater than MSP? A trivial metric that always just returns 1 would also be strictly greater than MSP. The analysis also oversteps; for example, the claim "Evidently, TRUST scores applied only to next-token prediction are generally the square of the MSP score in addition to higher order terms, but are always lower bounded by the square MSP." is false, since in reality you don't have a perfect judge.

**Questions:**

Could you clarify how the BERT approximation works? Is my understanding in the previous section correct?

---

> ### Author Response · Authors · 2025-11-20
>
> Please note that we have a general comment above that addresses the major points brought up by reviewers in a cohesive way. We encourage the reviewer to read that comment, but we also address individual reviewer comments below.
>
> Regarding the input and output of the BERT model, as well as other relevant implementation details, please refer to discussion point 3. in our general comment section. The BERT model essentially takes the sequence of tokens generated by the main LLM as input and it outputs predicted TRUST scores at each token position. It distills the uncertainty signal and reduces the time complexity of the TRUST algorithm at inference time from quadratic time to linear time, removing the need for generating full trial rollouts and for computing semantic similarity at inference time.
>
> Regarding the differences between TRUST and Semantic Entropy, please refer to discussion point 2. in our general comment section. We position TRUST as well-suited to long and causal real-world text datasets, while Semantic Entropy is only applicable to sentence-length Q&A datasets.
>
> Regarding the purpose of our toy synthetic dataset, please refer to discussion point 1. in our general comment section where we explain that this dataset is an example of aleatoric uncertainty. It is impossible to achieve consensus between distinct human preferences. In this dataset, the semantic similarity score helps detect the post-hoc strength of the aleatoric uncertainty signal, which we influenced pre-hoc through the dataset mixing ratio. On one hand, fine-tuned LLMs at lower mixing ratios produce the highest frequency response more frequently than the alternative responses. On the other hand, fine-tuned LLMs at higher mixing ratios produce more diverse responses.
>
> Regarding Theorem 1, to the extent of our knowledge, currently there is no tractable theoretical treatment that characterizes semantic spaces in multi-token settings. We offer the simple theoretical relation of TRUST with $MSP^2$ in Section 3.4 to show that we are on solid theoretical footing in appropriate limits, as MSP is an industry-standard measure of model uncertainty. Once we reach the multi-token limit, we must rely on semantic understanding within LLMs. We note that other multi-token uncertainty papers also do not provide much theoretical treatment, including the semantic uncertainty work by Kuhn et al [1].
>
> Also, we believe that the reviewer misinterpreted our treatment of Theorem 1. The point is not that TRUST is bounded below by $MSP^2$; the point is that TRUST is in fact equal to $MSP^2$ plus higher order terms. We therefore are making a stronger statement than the lower bound, but providing a proof of strong correlation to an industry-standard uncertainty measure. We can make this point clearer in a revision.
>
> [1] Kuhn, Lorenz, Yarin Gal, and Sebastian Farquhar. "Semantic uncertainty: Linguistic invariances for uncertainty estimation in natural language generation." arXiv preprint arXiv:2302.09664 (2023).

---

> > ### Comment · Reviewer_Bezx · 2025-11-24
> >
> > > Regarding the input and output of the BERT model, as well as other relevant implementation details, please refer to discussion point 3. in our general comment section. The BERT model essentially takes the sequence of tokens generated by the main LLM as input and it outputs predicted TRUST scores at each token position. It distills the uncertainty signal and reduces the time complexity of the TRUST algorithm at inference time from quadratic time to linear time, removing the need for generating full trial rollouts and for computing semantic similarity at inference time.
> >
> > Is there a reason the BERT model doesn't take embeddings from the parent model as input? I still think my point stands that I don't think the BERT model alone is going to be able to capture the uncertainty of the full model in more realistic settings. Also, I want to repeat my point that this is essentially what Kossen et al. 2024 already does, just with semantic entropy. Kossen is not cited in the current version of the manuscript.
> >
> > > Regarding the differences between TRUST and Semantic Entropy, please refer to discussion point 2. in our general comment section. We position TRUST as well-suited to long and causal real-world text datasets, while Semantic Entropy is only applicable to sentence-length Q&A datasets.
> >
> > I don't believe this speculation is responsive to the literature. Farquhar et al. 2024, the Nature version of the original semantic entropy paper, includes a long-form bio evaluation. That paper isn't mentioned here either.
> >
> > > Regarding the purpose of our toy synthetic dataset, please refer to discussion point 1. in our general comment section where we explain that this dataset is an example of aleatoric uncertainty. It is impossible to achieve consensus between distinct human preferences. In this dataset, the semantic similarity score helps detect the post-hoc strength of the aleatoric uncertainty signal, which we influenced pre-hoc through the dataset mixing ratio. On one hand, fine-tuned LLMs at lower mixing ratios produce the highest frequency response more frequently than the alternative responses. On the other hand, fine-tuned LLMs at higher mixing ratios produce more diverse responses.
> >
> > I'm not saying the uncertainty here isn't aleatoric, I'm saying it's the wrong kind of aleatoric uncertainty. You're not actually measuring "semantically-invariant dimensions such as synonyms or word choice" as claimed.
> >
> > > Also, we believe that the reviewer misinterpreted our treatment of Theorem 1. The point is not that TRUST is bounded below by $MSP^2$; the point is that TRUST is in fact equal to $MSP^2$ plus higher order terms. We therefore are making a stronger statement than the lower bound, but providing a proof of strong correlation to an industry-standard uncertainty measure. We can make this point clearer in a revision.
> >
> > You don't make the argument why these "higher-order terms" are the correct terms to include, or that they're beneficial at all.
> >
> > Given my outstanding objections and the fact that the revised version of the manuscript also fails to engage with any of the relevant literature (e.g. the sentence "Most commonly, LLM uncertainty still uses standard white-box statistical methods like entropy (Wang et al., 2022), max softmax (MSP) (Liu et al., 2023), or variations thereof (Kuhn et al., 2023)." is still in the related work section and grossly mischaracterizes Kuhn et al., 2023, by far the most relevant piece of related work), I'm sticking to my current score for now.

---

> > > ### Author Response · Authors · 2025-12-02
> > >
> > > > Is there a reason the BERT model doesn't take embeddings from the parent model as input? I still think my point stands that I don't think the BERT model alone is going to be able to capture the uncertainty of the full model in more realistic settings. Also, I want to repeat my point that this is essentially what Kossen et al. 2024 already does, just with semantic entropy. Kossen is not cited in the current version of the manuscript.
> > >
> > > There are a few reasons why semantic entropy probes as outlined in Kossen et al. [1] are lacking. First, the semantic entropy probes underperform the black-box semantic entropy method as seen in figures 1-4 from [1] and so our comparisons between TRUST and the black-box semantic entropy method supersede the method comparisons in [1]. Second, it is impractical to transfer all hidden states from all layers of a 70B model from GPU HBM memory to system RAM at inference time due to the well-known high latency of such transfers. Such transfers are required in [1] in order to run CPU-bound logistic regression probes in scikit-learn. Third, it is crucial that TRUST remains black-box compatible to engender adoption by practitioners.
> > >
> > > Again, we never claimed that the distilled BERT model is a novel contribution. We merely offered the distilled BERT model as an immediately practical method for production inference settings. We also disagree that the distilled BERT model achieves the same objective as [1], because our BERT model learns to predict granular token-level TRUST uncertainty signals that respect causal properties in complex text, while semantic entropy probes only learn to predict a limited sentence-level or few-token signal that disregards causality between sentences by design. Please see our general discussion point 2 above where we compare the Semantic Entropy method with the TRUST method and address this point quite explicitly.
> > >
> > > > I don't believe this speculation is responsive to the literature. Farquhar et al. 2024, the Nature version of the original semantic entropy paper, includes a long-form bio evaluation. That paper isn't mentioned here either.
> > >
> > > Please refer to discussion point 2 in the general comment where we address semantic entropy’s shortcomings. Note that the long-form bio dataset is a synthetic dataset of 20 biographies, and required **explicit factoid decomposition** for semantic entropy to work. TRUST removes this arbitrary structure by operating at the token level, thereby preserving causal dependencies while allowing restriction to sentence-level evaluation where needed. Design choices in semantic entropy such as entailment verification and factoid decomposition all limit the use of semantic entropy within general settings. Factoid decomposition in particular makes semantic uncertainty rather ill-suited towards organic real-world conversations, since proper entailment verification depends on relationships between interconnected sentences [2]. Empirically, TRUST strongly outperforms semantic entropy on the real-world MATH long-answer uncertainty prediction, which illustrates how semantic entropy can struggle on more complex tasks.
> > >
> > > > I'm not saying the uncertainty here isn't aleatoric, I'm saying it's the wrong kind of aleatoric uncertainty. You're not actually measuring "semantically-invariant dimensions such as synonyms or word choice" as claimed.
> > >
> > > We believe there is a misunderstanding here: we never claim that we are sensitive to the synonyms or word choice - in fact, quite the opposite. Synonyms and word choice perturbations should still register as semantically equivalent in our system, which would mean a measurement of *low* uncertainty. This is as intended.
> > >
> > > The type of aleatoric uncertainty in the toy example is precisely the type we want to be sensitive to: the type where there are multiple plausible answers to the same input but with different semantic branchings. The toy example encapsulates this type of uncertainty well, and TRUST performs admirably in quantifying when this type of aleatoric uncertainty is present.
> > >
> > > > You don't make the argument why these "higher-order terms" are the correct terms to include, or that they're beneficial at all.
> > >
> > > We are admittedly a bit confused by this comment. Higher order terms are meant to be ignored. We are saying that the primary effect is a linear correlation with MSP^2, in addition to other terms that can generally be ignored.
> > >
> > > [1] Kossen, J., Han, J., Razzak, M., Schut, L., Malik, S., & Gal, Y. (2024). Semantic Entropy Probes: Robust and Cheap Hallucination Detection in LLMs. arXiv. arXiv:2406.15927.
> > >
> > > [2] “Propositional Logic and SAT.” 15-281: Foundations of Intelligent Systems, Carnegie Mellon University, https://www.cs.cmu.edu/~15281-f23/coursenotes/proplogic/index.html.

---

### Official Review · Reviewer_AGJq · 2025-10-17

**Soundness:** 2
**Presentation:** 2
**Contribution:** 1
**Rating:** 2
**Confidence:** 4

**Summary:**

The paper proposes a method for measuring the uncertainty of LLMs in long-range text generations. The method relies on generating multiple samples from the LLM and computing their pairwise semantic similarity using a judge model. The average semantic similarity is then used as the uncertainty score. The authors show that their proposed measure has high correlation with a mixing factor on a preference dataset they propose, and that it can be predictive of the difficulty level on MATH questions.

**Strengths:**

- The work is tackling an important problem of uncertainty estimation in LLMs.
- The proposed TRUST scores operate in a black-box mode.

**Weaknesses:**

- **Significant related works are omitted.** This work seems to fit into the broad area of semantic uncertainty estimation in long-form LLM generations, which is not new. Yet, most related works are not discussed by the paper. In particular, semantic entropy proposed by [1, 2] feels like a more comprehensive version of the approach proposed in this paper, combining semantic clustering with entropy. Compared to semantic entropy, I do not see any methodological or empirical novelty provided by this submission.
- **Empirical evaluation is limited.** The paper does not compare to important baselines such as semantic entropy. Further, the evaluations are highly limited, non-standard and overall questionable. In particular:
    - Although the TRUST scores are presented as an uncertainty metric, no calibration results are provided, showing whether the proposed TRUST scores can be reliably used to predict model’s accuracy.
    - The preference dataset used in Sec.  4.2 seems arbitrary and not particularly suited for evaluation of uncertainty metrics. For starters, it is unclear to me what would be the value of quantifying uncertainty on a preference dataset? (In this case there is no notion of “accuracy”, so the uncertainty score cannot really be used to improve the reliability of the model). That makes the evaluation of the any proposed uncertainty metric on this dataset very difficult. Secondly, why is correlation with the mixing factor a desirable property? What is the purpose of fine-tuning of the model on this dataset?
    - On the MATH dataset, the TRUST scores are not used to predict the accuracy of the model, but rather the difficulty level of the question. While the model accuracy can be related to task difficulty, the two are not equivalent. It seems to me that what the experiments in section 4.3 are doing is showing that more difficult tasks lead to more diverse generations (which makes intuitive sense, as in maths problems there is usually more diverse ways to reach a conclusion on a more complex task). However, the paper does not in any way address whether the TRUST scores can be used to predict the accuracy of the model.
- **Lacking implementation details and ablations.** In section 3.3 the authors propose to estimate the TRUST scores using a predictive model. However, no relevant details about the training of the said predictive models are provided, deeming the results not reproducible. Further, the authors do not ablate the effect of the number of generations N, or whether pairwise vs sequential comparisons are more efficient.
- **Theory does not seem relevant.** In section 3.4 the authors provide a theoretical results meant to validate their proposed TRUST score. However, the result is based on an extremely simplified setting (single-step generation, ideal judge, binary 0-1 label), making this theoretical aspect a dubious contribution. Furthermore, it is unclear why being lower bounded by squared MSP is a desirable property.


[1] https://arxiv.org/pdf/2302.09664
[2] https://www.nature.com/articles/s41586-024-07421-0

**Questions:**

- What is meant by “first possible response” in line 315? What’s the rationale behind using the mixing factor in the preference dataset? How do the datasets obtained with the different mixing factors actually differ?
- What is the training procedure for the TRUST predictive model? What dataset was used? Was a linear model used?

---

> ### Author Response · Authors · 2025-11-20
>
> Please note that we have a general comment above that addresses the major points brought up by reviewers in a cohesive way. We encourage the reviewer to read that comment, but we also address individual reviewer comments below.
>
> Regarding the related work of Semantic Entropy, please refer to our discussion point 2. in the general comments section, where we disagree that Semantic Entropy is a more comprehensive method than TRUST. The TRUST method operates at token-level and is agnostic to complexities such as causal relations inside multi-sentence outputs of organic real-world datasets, while Semantic Entropy limits its applicability to simple outputs through design choices like entailment verification and factoid decomposition which fail to capture semantic dependencies between sentences on the more complex MATH dataset.
>
> Both semantic entropy [1,2] and our TRUST scores perform at parity on our toy preferences dataset when we inspect the correlation coefficient with respect to the mixing ratio, which is expected in such a simple setting:
> | Model                  | TRUST correlation | Semantic Entropy correlation |
> |-------------------------|------------|------------------------|
> | fine-tuned Llama-3.1-8B | 0.9756 ± 0.0013 | 0.9764 ± 0.0013 |
>
> Compared to semantic entropy [1,2], which decomposes factual accuracy into atomic factoids, our broader setting with the MATH dataset contains causal dependencies that are not decomposable. Still even without factoid decomposition, our TRUST scores strongly outperform semantic entropy on MATH as in the table below. We hypothesize that semantic entropy is not very well-suited for longer outputs, while TRUST better captures model uncertainty in complex scenarios while leaving base accuracy unchanged.
> | Model         | TRUST MSE | Semantic Entropy MSE |
> |----------------|------------|----------------------|
> | GPT-4o-mini    | 1.2042 ± 0.0615 | 1.5614 ± 0.0558 |
> | Llama-3.1-70B  | 1.2193 ± 0.0219 | 1.4368 ± 0.0498 |
>
> Regarding the value of the proposed uncertainty targets, we refer to our discussion point 1. in the general comments section. We intentionally avoided calibrating towards accuracy measures since accuracy is orthogonal to uncertainty. Instead we calibrated towards proxy measures of uncertainty such as the mixing ratio of our aleatoric uncertainty toy dataset and the math problem difficulty level of the epistemic uncertainty MATH dataset.
>
> To clarify what "first possible response" means on line 315, note that we provided our toy dataset as supplementary material for the initial submission. For example the initial 10 rows pertain to the question "What's your favorite pet?" and there are 10 responses, the first of which is "I prefer parrots because they are colorful and can mimic sounds" and that is the "first possible response" for this question. The choice of a dominant response in the fine-tuning procedure can be arbitrary. However, the frequency of this dominant response relative to the other responses for the same question (also referred to as the "mixing ratio") in the resampled training dataset is what gives us control to inject an aleatoric uncertainty signal.
>
> Regarding further implementation details of our BERT TRUST predictive model, please refer to our discussion point 3. in the general comments section. The basic TRUST algorithm generates multiple full trial rollouts at each token position and computes pair-wise semantic similarity, so the time complexity is $O(k^2)$. The BERT model distills the TRUST semantic similarity score at each token position, which removes the need for generating full trial rollouts and semantic similarity scores at inference time, therefore reducing its time complexity to $O(k)$. Furthermore, we implemented disjoint pair-wise semantic comparisons to ensure IIDness of samples when averaging the TRUST score, which is more efficient than comparing all trial rollouts to each other.
>
> Regarding Theorem 1, to the extent of our knowledge, currently there is no tractable theoretical treatment that characterizes semantic spaces in multi-token settings. We offer the simple theoretical relation of TRUST with $MSP^2$ in Section 3.4 to show that we are on solid theoretical footing in appropriate limits, as MSP is an industry-standard measure of model uncertainty. Once we reach the multi-token limit, we must rely on semantic understanding within LLMs. We note that other multi-token uncertainty papers also do not provide much theoretical treatment, including the semantic uncertainty work by Kuhn et al [1].
>
> [1] Kuhn, Lorenz, Yarin Gal, and Sebastian Farquhar. "Semantic uncertainty: Linguistic invariances for uncertainty estimation in natural language generation." arXiv preprint arXiv:2302.09664 (2023).
>
> [2] Farquhar, S., Kossen, J., Kuhn, L. et al. Detecting hallucinations in large language models using semantic entropy. Nature 630, 625–630 (2024). https://doi.org/10.1038/s41586-024-07421-0.

---

> > ### Comment · Reviewer_AGJq · 2025-11-27
> >
> > After reading the authors’ responses I think that there is a lot of ambiguity around the actual objective of this work. The authors argue that they care purely about quantifying the “uncertainty” of LLMs, which (according to them) is orthogonal to accuracy. I believe that when saying “uncertainty” the authors really mean “diversity of generations”, as opposed to “confidence”. To be precise, for me the notion of confidence is closely related to the concept of confidence sets: if the model is 90% confident, its answers are correct 90% of the time, on average. Thus, confidence is by definition related to accuracy.
> >
> > In the rebuttals the authors vehemently claim that their work is orthogonal to accuracy (and hence by proxy, also to confidence). However, this is not evident from the way that the paper is written. Some of the related works mentioned in the introduction address “uncertainty” strictly in the confidence sense that I describe above, analysing calibration and hallucinations (e.g. Xiong et al. 2023, Lin et al. 2022, Kalai et al. 2025). If this work indeed intends to address uncertainty purely from the generation diversity perspective, I would expect this to be well-motivated and explained in the introduction and clarified in the related works section. In particular, I would want to see a detailed case for why caring purely for diversity of generations (rather than for confidence) matters and what are its use cases. Further, given that the current use of the word “uncertainty” is ambiguous, the authors’ objective should be clearly stated and defined early on in the paper.
> >
> > Regardless of this discussion and ambiguity, I still think that this paper is of low quality and does not meet scientific standards.
> >
> > I still do not understand how the preference datasets are constructed. Could the authors clarify how many samples are in each of the resampled datasets? That is, how many times does each question appear in the training dataset? My current understanding is that each question appears only once, and its answer is chosen to be the “first” answer $(100 - 10M)%$ of the times, and a random other answer the remaining times. However, if this is the case, I do not see how the answer to one question is supposed to affect the answer to any other questions, as they are semantically unrelated (a point also raised by Reviewer Bezx).
> >
> > Independent of the resampling strategy, I maintain that drawing significant scientific conclusions based on fine-tuning an 8B parameter model using a "toy" dataset of only 80 unique questions is methodologically unsound. The risk of overfitting is high, and the generalisability of these results is questionable.
> >
> > Regarding the MSE calculation on the MATH dataset, I still do not understand why being able to predict the difficulty of the question from the TRUST score is a desirable property. As I explained above, I think that what the authors are really showing is that more difficult questions tend to have more _diverse_ generations. While this is a somewhat interesting observation, I do not think though that this result in any way validates TRUST as a useful metric. Even if the authors decided to pivot their paper towards "diversity quantification", I am not convinced that difficulty in the MATH dataset is a good proxy for diversity.
> >
> > Finally, I stand by my point that the theoretical results presented in the paper are not relevant. It remains unclear why being lower bounded by MSP is a desirable property (as pointed out by reviewer Bezx, any metric which has a fixed value of 1 also achieves this).

---

> > > ### Author Response · Authors · 2025-12-02
> > >
> > > > After reading the authors’ responses I think that there is a lot of ambiguity around the actual objective of this work. The authors argue that they care purely about quantifying the “uncertainty” of LLMs, which (according to them) is orthogonal to accuracy. I believe that when saying “uncertainty” the authors really mean “diversity of generations”, as opposed to “confidence”. To be precise, for me the notion of confidence is closely related to the concept of confidence sets: if the model is 90% confident, its answers are correct 90% of the time, on average. Thus, confidence is by definition related to accuracy.
> > >
> > > We firmly disagree with the comment that accuracy and confidence should be conflated together. Please see our discussion point 1 in the general comment that we posted. We believe the reviewer is confusing “confidence” with “calibration.” This conflation would be acceptable if the model we are querying was well-calibrated, but we note that this is almost always not the case, and especially not so for off-the-shelf LLMs! These models are known to be confidently incorrect, for example, which reveals a lack of calibration and thus a breakage of the relationship between calibration/accuracy and confidence.
> > >
> > > TRUST measures uncertainty in the sense of the spread of predictions for each example in the dataset, which is completely separated from accuracy measurements as it does not reference ground truth. We have generally observed that predicting *accuracy* is different from predicting *label variance*, and the latter is what TRUST measures (which sets it apart from many other methods like semantic entropy). We are working on a grounded study of why accuracy is different from confidence, and hope to include these studies in the manuscript soon.
> > >
> > > > I still do not understand how the preference datasets are constructed. Could the authors clarify how many samples are in each of the resampled datasets? That is, how many times does each question appear in the training dataset? My current understanding is that each question appears only once, and its answer is chosen to be the “first” answer  of the times, and a random other answer the remaining times. However, if this is the case, I do not see how the answer to one question is supposed to affect the answer to any other questions, as they are semantically unrelated (a point also raised by Reviewer Bezx).
> > >
> > > Here we clarify how the resampled preference datasets are constructed.
> > > - Mixing ratio = 0: For each of the 80 questions there are 10 datapoints that contain the question with the same duplicated response for a total of 800 datapoints. This has the lowest aleatoric uncertainty.
> > > - Mixing ratio = 1: For each question there are 9 datapoints that contain the same duplicated response as well as 1 datapoint that contains another different response.
> > > - Mixing ratio = 2: For each question there are 8 datapoints that contain the same duplicated response as well as 2 datapoints that contain 2 other different responses.
> > > …
> > > - Mixing ratio = 9: For each question there are no duplicated responses, but there are 10 distinct datapoints that contain 10 distinct responses. This has the highest aleatoric uncertainty.
> > >
> > > The mixing ratio of responses in the resampled datasets is learnable and represents the irreducible aleatoric uncertainty signal, which increases with the increased ambiguity of responses at high mixing ratios. The mixing ratios are therefore a clear measure of aleatoric uncertainty.
> > >
> > > > Independent of the resampling strategy, I maintain that drawing significant scientific conclusions based on fine-tuning an 8B parameter model using a "toy" dataset of only 80 unique questions is methodologically unsound. The risk of overfitting is high, and the generalisability of these results is questionable.
> > >
> > > The resampled datasets contain 800 datapoints each, since there are 80 questions with 10 responses each. In practice, we find this to be an adequate dataset scale to successfully fine-tune a medium-sized 8B parameter model’s behavior and prove that this model can recover the aleatoric mixing ratio signal at inference time.
> > >
> > > We note that if we had observed overfitting, then we would have seen our fine-tuned models mode collapse into a majority classifier, which was not the case in practice. Overfitting would have resulted in *materially worse performance* of our TRUST scores, but we instead see a strong correlation between mixing ratio and TRUST. This result is indicative that we are far from the overfitting region.
> > >
> > > In general, aleatoric uncertainty due to hard conflicting examples in the training dataset is relevant to real-world settings where label impurity in pre-training datasets can cause divergent and conflicting LLM responses at inference time ultimately contributing to the problem of LLM unreliability. Detecting this impurity signal through sampling methods such as TRUST is quite useful.

---

### Official Review · Reviewer_h7Et · 2025-10-26

**Soundness:** 2
**Presentation:** 3
**Contribution:** 2
**Rating:** 4
**Confidence:** 4

**Summary:**

This paper proposes TRUST (Temperature-Related Unambiguity via Similarity Tracking), a black-box method for estimating uncertainty in large language models. Instead of relying on token-level entropy or max-softmax probabilities (MSP), TRUST quantifies semantic uncertainty by generating multiple completions from a model at a fixed non-zero temperature and measuring their pairwise semantic similarity using a “judge” LLM. High similarity implies low uncertainty. In parallel, a lightweight BERT mode is trained to predict these TRUST scores efficiently. The paper shows a theoretical link between TRUST and squared MSP in the single-token limit and evaluates the method on a small synthetic dataset and the MATH benchmark, where TRUST (and BERT estimation version) outperform entropy, MSP, and ensemble-based baselines.

**Strengths:**

1. Using semantic similarity across temperature-sampled completions for uncertainty estimation is intuitive and seems to be new.

2. Works in a pure black-box setting, requiring no access to logits or gradients.

3. Simple relationship to MSP gives theoretical grounding.

4. Clear writing and easy to follow.

**Weaknesses:**

1. Only two datasets; no downstream tasks (e.g., calibration or hallucination detection). More experiments can be more convincing of the results and it should not be a lot of efforts.

2. Sensitivity to the choice of the “judge” LLM and maybe to prompts too, perhaps some variants and analysis is needed.

3. Requires multiple generations per input; distillation helps but need to be tested properly. Some follow up experiments are needed.

4. Not sure how Theorem 1 generalise to vary length output and beyond single token cases.

**Questions:**

Please see my weakness section for more information.

---

> ### Author Response · Authors · 2025-11-20
>
> Please note that we have a general comment above that addresses the major points brought up by reviewers in a cohesive way. We encourage the reviewer to read that comment, but we also address individual reviewer comments below.
>
> We appreciate the reviewer's suggestion to run more experiments with different judge LLMs, different judge prompts, and to show how the BERT model extends beyond the chosen validation set on the MATH dataset. We will run those experiments in time for the camera-ready if allowed to continue.
>
> However, we predict that the results will not change significantly. We have indeed tested judge prompts with the gpt-4, gpt-4o, gpt-4.1, gpt-5 series as well as llama-3.1-8b, llama-3.1-70b, llama-3.3-70b in a smaller sample of the MATH dataset, but we found that any one of these versions provides similarly calibrated performance for the semantic similarity task. We will include details from some of these experiments in a revision.
>
> Here we show how changing the LLM judge model but using the same system prompt doesn't influence the frequency of 1-5 similarity scores significantly over the last 100 examples of the MATH dataset.
>
> | Model / Score |     1 |     2 |     3 |     4 |     5 |
> |:--------------|------:|------:|------:|------:|------:|
> | gpt-4-turbo   | 0.144 | 0.336 | 0.074 | 0.101 | 0.345 |
> | gpt-4.1       | 0.142 | 0.342 | 0.07  | 0.1   | 0.346 |
> | gpt-4.1-mini  | 0.142 | 0.335 | 0.072 | 0.101 | 0.35  |
> | gpt-4.1-nano  | 0.148 | 0.334 | 0.069 | 0.104 | 0.345 |
> | gpt-4o        | 0.142 | 0.345 | 0.069 | 0.097 | 0.347 |
> | gpt-4o-mini   | 0.146 | 0.334 | 0.075 | 0.098 | 0.347 |
> | gpt-5         | 0.143 | 0.337 | 0.075 | 0.099 | 0.346 |
> | gpt-5-mini    | 0.138 | 0.345 | 0.068 | 0.104 | 0.345 |
> | gpt-5-nano    | 0.147 | 0.329 | 0.081 | 0.094 | 0.349 |
> | llama-3.1-70b | 0.143 | 0.336 | 0.074 | 0.098 | 0.349 |
> | llama-3.1-8b  | 0.148 | 0.333 | 0.078 | 0.093 | 0.348 |
> | llama-3.3-70b | 0.144 | 0.335 | 0.077 | 0.093 | 0.351 |
>
> Here we show how increasing the number of trial rollouts with fixed gpt-4o judge model and fixed system prompt above the current value of N = 10 doesn't influence the frequency of 1-5 similarity scores significantly over the last 100 examples of the MATH dataset.
>
> |   Trial / Score |     1 |     2 |     3 |     4 |     5 |
> |--------:|------:|------:|------:|------:|------:|
> |       5 | 0.132 | 0.338 | 0.064 | 0.1   | 0.366 |
> |      10 | 0.142 | 0.345 | 0.069 | 0.097 | 0.347 |
> |      15 | 0.137 | 0.365 | 0.061 | 0.097 | 0.341 |
> |      20 | 0.138 | 0.358 | 0.061 | 0.098 | 0.346 |
> |      25 | 0.132 | 0.355 | 0.061 | 0.095 | 0.357 |
> |      30 | 0.129 | 0.357 | 0.061 | 0.096 | 0.357 |
> |      35 | 0.133 | 0.352 | 0.063 | 0.097 | 0.355 |
> |      40 | 0.133 | 0.35  | 0.063 | 0.097 | 0.357 |
> |      45 | 0.132 | 0.348 | 0.067 | 0.096 | 0.357 |
> |      50 | 0.129 | 0.348 | 0.067 | 0.097 | 0.359 |
>
> Regarding Theorem 1, to the extent of our knowledge, currently there is no satisfactory theoretical treatment that characterizes semantic spaces in multi-token settings. We offer the simple theoretical relation of TRUST with $MSP^2$ in Section 3.4 to show that we are on solid theoretical footing in appropriate limits, but we rely on semantic understanding within LLMs for longer token lengths. In our understanding this tactic is fairly industry standard and theory in the multi-token limit would be rather intractable.
>
> We explicitly decided to not tackle hallucination detection because we wanted to focus our setting on *uncertainty* rather than *accuracy*. Although these two concepts are related, the relationship is usually coincidental and hallucinations are based on a “ground truth” reference which is grounded in factual information. In our case, we created settings that are purely tuned for aleatoric uncertainty (our toy preference dataset, where the uncertainty arises due to irreducible dataset ambiguity) and epistemic uncertainty (the MATH dataset, where the uncertainty arises due to intrinsic problem difficulty but each problem is well-defined and learnable). In this way we feel TRUST is a much more precise tackling of uncertainty estimation than accuracy-based metrics which tend to conflate correctness with confidence. If interested, please see our general comment above for more discussion.

---

> > ### Comment · Reviewer_h7Et · 2025-11-23
> > **Response to Authors**
> >
> > I am generally happy with the further experiments for both more judge models and many trials. However, I strongly disagree with the points the authors suggest that accuracy (or performance in general) is not important for uncertainty quantification. Considering other reviewers' opinion, I think 4 is a very fair score for this paper.

---

> > > ### Author Response · Authors · 2025-11-25
> > >
> > > We thank you for your response and are glad you appreciated our additional experiments. We want to clarify that our viewpoint is not that accuracy is completely uncorrelated from difficulty/ambiguity as an uncertainty signal, but is just less direct. Our further experiments do show that methods like semantic uncertainty which are validated on accuracy measures struggle with uncertainty metrics like MATH difficulty, which warrants some discussion around whether accuracy is capturing the most useful uncertainty signal. We would certainly not claim that accuracy is uncorrelated with uncertainty, but do feel that it is more of a convenient proxy rather than a direct measurement of the signal we want. If the reviewer would let us know why they feel otherwise, we would be happy to engage in that discussion, especially as we feel this topic lies at the very core of uncertainty prediction as a subfield.

---

### Official Review · Reviewer_7xAU · 2025-10-31

**Soundness:** 1
**Presentation:** 1
**Contribution:** 1
**Rating:** 2
**Confidence:** 5

**Summary:**

This paper introduces a method for estimating LLM uncertainty by sampling multiple outputs at nonzero temperature and computing the pairwise semantic similarity between outputs using another LLM as a “judge.” The authors claim that TRUST offers a black-box, semantically aware measure of uncertainty and that it outperforms conventional baselines such as MSP and entropy. Experiments are conducted on a toy “preference” dataset and the MATH dataset, where TRUST is used to predict task difficulty.

**Strengths:**

- The paper is written in readable English and the experimental setup is described in reasonable detail.
- The idea of leveraging temperature-induced sampling variance is intuitive (though not new).

**Weaknesses:**

1. **No technical contributions.** TRUST is just a rephrasing of “semantic similarity among multiple LLM samples” , a concept already well explored in the literature. For example, Semantic Uncertainty [1]. The authors do not identify or address any new technical challenge. The theoretical section merely shows that TRUST $\approx$ (MSP)², which is trivial and contributes nothing conceptually. The baselines are also very weak. Only entropy, MSP, and ensembling are compared. The paper completely ignores more relevant uncertainty or calibration methods (semantic uncertainty, conformal prediction, calibration-based OOD detection, etc.). The claimed “superiority” is therefore meaningless.
2. **Unconvincing experiments.** The “simple uncertainty prediction” task is an small-scale, synthesized, unverified dataset of random preferences. It has nothing to do with real uncertainty estimation. The “difficulty prediction” experiment on MATH is also questionable: predicting problem difficulty is not the same as predicting model uncertainty. None of the evaluations measure whether TRUST correlates with correctness or epistemic uncertainty.
3. **No link between uncertainty and accuracy.** The authors explicitly state they “did not consider model accuracy” when evaluating uncertainty, which undermines the entire motivation. Measuring uncertainty without checking calibration or correlation with correctness is meaningless.
4. **Poor presentation and professionalism.** Figures are low-resolution raster images rather than vector graphics. Many equations occupy large space for trivial content, such as Eq. 1 and 2. The writing often reads like a blog post rather than a research paper. The overall presentation does not meet the standards expected of ICLR submissions.
5. **No insight or takeaways.** The paper neither advances understanding of uncertainty in LLMs nor introduces any practical or theoretical framework that others can build upon. It is an incremental, poorly justified application of existing ideas.

[1] Kuhn, Lorenz, Yarin Gal, and Sebastian Farquhar. "Semantic uncertainty: Linguistic invariances for uncertainty estimation in natural language generation." arXiv preprint arXiv:2302.09664 (2023).

**Questions:**

Please refer to the weakness.

---

> ### Author Response · Authors · 2025-11-20
>
> Please note that we have a general comment above that addresses the major points brought up by reviewers in a cohesive way. We encourage the reviewer to read that comment, but we also address individual reviewer comments below.
>
> We appreciate the reviewer's concern related to 2. Unconvincing experiments and 3. No link between uncertainty and accuracy. Please refer to our discussion point 1. in the general comments section, where we disagree with the claim that model accuracy must be evaluated alongside uncertainty. Accuracy is orthogonal over uncertainty. Our work explicitly models aleatoric and epistemic uncertainty signals, not accuracy as a proxy for uncertainty.
>
> We also appreciate the reviewer's concern related to 1. No technical contributions. Please refer to our discussion point 2. in the general comments section where we discuss the differences between the TRUST method and the Semantic Entropy method. The TRUST method operates at token-level and is agnostic to complexities such as causal relations inside multi-sentence outputs of organic real-world datasets, while Semantic Entropy limits its applicability to simple outputs through design choices like entailment verification and factoid decomposition which fail to capture semantic dependencies between sentences on the more complex MATH dataset.
>
> Both semantic entropy [1,2] and our TRUST scores perform at parity on our toy preferences dataset when we inspect the correlation coefficient with respect to the mixing ratio, which is expected in such a simple setting:
> | Model                  | TRUST correlation | Semantic Entropy correlation |
> |-------------------------|------------|------------------------|
> | fine-tuned Llama-3.1-8B | 0.9756 ± 0.0013 | 0.9764 ± 0.0013 |
>
> Compared to semantic entropy [1,2], which decomposes factual accuracy into atomic factoids, our broader setting with the MATH dataset contains causal dependencies that are not decomposable. Still even without factoid decomposition, our TRUST scores strongly outperform semantic entropy on MATH as in the table below. We hypothesize that semantic entropy is not very well-suited for longer outputs, while TRUST better captures model uncertainty in complex scenarios while leaving base accuracy unchanged.
> | Model         | TRUST MSE | Semantic Entropy MSE |
> |----------------|------------|----------------------|
> | GPT-4o-mini    | 1.2042 ± 0.0615 | 1.5614 ± 0.0558 |
> | Llama-3.1-70B  | 1.2193 ± 0.0219 | 1.4368 ± 0.0498 |
>
> Regarding the comment on poor presentation and professionalism, we note that we find our graphics to represent appropriately what we want and that they are very readable - if there are any examples of graphics where the reviewer had trouble reading any text please let us know and we will offer a fix. Otherwise, we would appreciate specific examples of the other comments around writing style that the reviewer found at issue.
>
> [1] Kuhn, Lorenz, Yarin Gal, and Sebastian Farquhar. "Semantic uncertainty: Linguistic invariances for uncertainty estimation in natural language generation." arXiv preprint arXiv:2302.09664 (2023).
>
> [2] Farquhar, S., Kossen, J., Kuhn, L. et al. Detecting hallucinations in large language models using semantic entropy. Nature 630, 625–630 (2024). https://doi.org/10.1038/s41586-024-07421-0.

---

### Official Review · Reviewer_Q3NW · 2025-11-01

**Soundness:** 2
**Presentation:** 2
**Contribution:** 2
**Rating:** 2
**Confidence:** 3

**Summary:**

This paper proposes TRUST (Temperature-Related Unambiguity via Similarity Tracking), a black-box method for estimating uncertainty in LLMs. Instead of relying on logits or token-level entropy, TRUST samples multiple completions from the same prompt at nonzero temperature and uses a separate judge LLM to compute pairwise semantic similarity among outputs. The average similarity serves as an inverse measure of uncertainty: higher similarity means lower uncertainty. The authors show a theoretical link between TRUST and squared Maximum Softmax Probability (MSP) in the single-token limit and claim strong empirical results on both a synthetic preference dataset and the MATH benchmark for difficulty prediction. A distilled BERT model trained to approximate TRUST achieves similar performance with lower inference cost.

**Strengths:**

+ The proposed uncertainty score requires no access to logits or model internals, making it deployable with closed APIs.
+ Outperforms entropy, MSP, and even ensemble-based metrics in tasks such as MATH difficulty prediction.

**Weaknesses:**

- The theoretical depth is limited only to a one-token, idealized setting, while multi-token behavior is purely empirical. The roles of temperature, number of samples, and judge prompt format are heuristic without principled justification.
- Generating multiple completions and pairwise similarity judgments may be expensive in computational cost. The paper did not conduct a complexity analysis.
- The use of pairwise semantic distances and averaging is intuitive but lacks a clear probabilistic interpretation or calibration guarantee.
- Benchmarks are small (a toy preference set and MATH).

**Questions:**

Can the authors demonstrate why the output text of LLMs will have natural sampling variance/uncertainty given a nonzero temperature?

---

> ### Author Response · Authors · 2025-11-19
>
> Please note that we have a general comment above that addresses the major points brought up by reviewers in a cohesive way. We encourage the reviewer to read that comment, but we also address individual reviewer comments below.
>
> Regarding algorithm complexity analysis and computational cost:
> - Please refer to discussion point 3. in the general comment section where we discuss time complexity.
> - The basic TRUST algorithm has time complexity $O(k^2)$ for a response length of k tokens, dominated by generating full rollouts at each token position.
> - The distilled BERT model reduces time complexity at inference time to $O(k)$ since it no longer requires generating full trial rollouts and pair-wise comparisons at each token position.
> - The rollout generation with gpt-4o-mini and the semantic similarity comparisons with gpt-4o for the 300 MATH examples costs under \$100 and takes ~3 days with 30 parallel workers.
>
> Regarding the lack of a clear probabilistic interpretation for pairwise semantic distances and averaging in multi-token settings:
> - To the extent of our knowledge, currently there is no satisfactory theoretical treatment that characterizes semantic spaces in multi-token settings. However, LLMs have proven to be successful in tasks that require semantic understanding, and are generally considered state-of-the-art. We offer the simple theoretical relation of TRUST with $MSP^2$ in Section 3.4 and we show that TRUST is empirically correlated with $MSP^2$ in Figure 2 and Table 1 as part of the toy preferences experiment.
> - LLM-as-a-judge is useful as an approximate labeling method. Otherwise it would be very expensive to obtain these labels from a pool of human judges given that we need to complete comparisons for every token position and for every example. Relying on the LLM's natural semantic understanding is currently industry standard in these cases.
> - Using pairwise semantic similarity preserves the IIDness of samples when averaging. We make no calibration guarantees because all our results are based on *relative* TRUST measurements, which removes the necessity of absolute calibration.
>
> Regarding the size of the benchmarks being small:
> - Indeed our toy preference dataset consists of only 80 examples, but the effect size is extremely large given that we control the dataset generation down to the example-level.
> - The MATH dataset also contains 300 examples but they are highly complex with copious amounts of output. The effect size we showed with TRUST here is quite large, as you can visibly see in Figure 4.
>
> Why is there natural sampling variance at a nonzero temperature?
> - Nondeterminism within LLMs at nonzero temperature is a well-known phenomenon, and arises from the fact that logit probabilities are scaled by 1/temperature before being sent into the softmax function. Sampling from the softmax distribution then creates natural variability in the output.
> - We may be misinterpreting the reviewer’s comments here, so please let us know if there is anything specific within this mechanism that the reviewer would like us to clarify.
> - Beyond this natural sampling mechanism, there are other sources of randomness that [1] uncovered recently, but this is not the object of our study.
>
> [1] He, Horace and Thinking Machines Lab, "Defeating Nondeterminism in LLM Inference",
> Thinking Machines Lab: Connectionism, Sep 2025.

---

### Author Response · Authors · 2025-11-19
**General comments about accuracy, semantic uncertainty and further implementation details**

# 1. Why don't we evaluate model accuracy?
- We disagree with the claim that model accuracy must be evaluated alongside uncertainty. Accuracy measures average proximity to truth while uncertainty measures spread as shown in [3]. They are orthogonal to each other. Our work explicitly isolates uncertainty signals (aleatoric and epistemic) rather than using accuracy as a proxy. We avoided hallucination or factuality benchmarks precisely because those reflect accuracy, not uncertainty.
- Our calibration targets capture distinct uncertainty types: (1) aleatoric uncertainty in a preference dataset, where the uncertainty signal arises due to irreducible dataset ambiguity and is tunable via the mixing ratio; and (2) epistemic uncertainty in MATH, where all input/outputs are well-defined and learnable but are separated purely by difficulty.
- Both semantic entropy [1,2] and our TRUST scores perform at parity on our toy preferences dataset when we inspect the correlation coefficient with respect to the mixing ratio, which is expected in such a simple setting:
| Model                  | TRUST correlation | Semantic Entropy correlation |
|-------------------------|------------|------------------------|
| fine-tuned Llama-3.1-8B | 0.9756 ± 0.0013 | 0.9764 ± 0.0013 |
- Compared to semantic entropy [1,2], which decomposes factual accuracy into atomic factoids, our broader setting with the MATH dataset contains causal dependencies that are not decomposable. Still even without factoid decomposition, our TRUST scores strongly outperform semantic entropy on MATH as in the table below. We hypothesize that semantic entropy is not very well-suited for longer outputs, while TRUST better captures model uncertainty in complex scenarios while leaving base accuracy unchanged.
| Model         | TRUST MSE | Semantic Entropy MSE |
|----------------|------------|----------------------|
| GPT-4o-mini    | 1.2042 ± 0.0615 | 1.5614 ± 0.0558 |
| Llama-3.1-70B  | 1.2193 ± 0.0219 | 1.4368 ± 0.0498 |

# 2. Is Semantic Entropy more comprehensive than TRUST?
We disagree that Semantic Entropy [1,2] is more comprehensive than TRUST. TRUST is a broader, setting-agnostic uncertainty measure.
Semantic Entropy assumes independence between sentences, breaking causal links within paragraphs. TRUST operates at the token level, preserving causal dependencies while allowing restriction to sentence-level evaluation where needed. The other design choices in semantic entropy such as entailment verification and factoid decomposition all limit the use of semantic entropy within general settings. Factoid decomposition in particular makes semantic uncertainty rather ill-suited towards organic real-world conversations, since proper entailment verification depends on relationships between interconnected sentences [4].
Empirically, TRUST strongly outperforms semantic entropy on MATH long-answer uncertainty prediction, which illustrates how semantic entropy can struggle on more complex tasks.

# 3. What are the implementation details of TRUST?
- The basic TRUST algorithm has time complexity $O(k^2)$ for a response length of k tokens, dominated by generating full rollouts at each token position. For each question, we progressively fix ground-truth response tokens and generate full rollouts; the number of trial rollouts per step is a constant 10 which does not affect asymptotic complexity.
- We also train a BERT-based TRUST predictor for 40 epochs in the case of gpt-4o-mini rollouts, for 20 epochs in the case of llama-3.1-70b rollouts and for 50 epochs in the case of llama-3.1-8b rollouts. Once trained, it estimates uncertainty per token in a single forward pass without requiring any trial rollouts, thereby reducing inference complexity to $O(k)$.
- For the MATH dataset, the BERT model takes tokenized LLM completions as input and predicts basic TRUST scores learned via regression with a linear output layer (detailed model structure in Appendix A.5).
- For the toy preference dataset, to build the generation model we fine-tuned llama-3.1-8b with LoRA adapters and then generated rollouts as well as basic TRUST scores. Finally, we trained a BERT model with the same approach as for the MATH dataset.

[1] Kuhn, Lorenz, Yarin Gal, and Sebastian Farquhar. "Semantic uncertainty: Linguistic invariances for uncertainty estimation in natural language generation." arXiv preprint arXiv:2302.09664 (2023).

[2] Farquhar, S., Kossen, J., Kuhn, L. et al. Detecting hallucinations in large language models using semantic entropy. Nature 630, 625–630 (2024). https://doi.org/10.1038/s41586-024-07421-0.

[3] Wikipedia contributors. (2025, November 3). Accuracy and precision. In Wikipedia, The Free Encyclopedia. https://en.wikipedia.org/wiki/Accuracy_and_precision.

[4] “Propositional Logic and SAT.” 15-281: Foundations of Intelligent Systems, Carnegie Mellon University, https://www.cs.cmu.edu/~15281-f23/coursenotes/proplogic/index.html.

---

### Author Response · Authors · 2025-12-03

We want to thank all reviewers and ACs for taking the time to review our work. We noticed three central themes within the reviews: the relationship between accuracy and uncertainty, a comparison between TRUST and Semantic Entropy from existing literature, and the implementation details of our distilled BERT model. We updated our paper with multiple additional experiments to show that TRUST outperforms semantic entropy, and also added discussion both in the paper revision and within our OpenReview responses around why we chose to directly target uncertainty rather than using model accuracy as a convenient proxy. We added extensive review of related work within our revision and of semantic entropy in particular; our baselines show that semantic entropy (as expected) struggles with long-form complex responses and its attempts to work around this obstacle (e.g. factoid decomposition) tend to underperform in truly organic real-world settings. TRUST does not suffer from the same shortcomings. We also clarified the implementation of the basic TRUST algorithm and of the distilled BERT model, which allows TRUST to be immediately practical as a purely black-box method for uncertainty detection in production inference settings.

It is unfortunate that due to the recent data breach, we are unable to continue our discussion with the reviewers. However, we hope the reasonable observer agrees that we addressed all outstanding concerns with both convincing additional experimental evidence as well as clear supplementary exposition. We believe that TRUST fills a significant gap in practical, blackbox detection of uncertainty in long-form complex real-world datasets, which will be highly appreciated by both AI practitioners and researchers.

---

### Meta-Review · Area_Chair_k62E · 2026-01-06

**Summary:**

This paper introduces a black-box method for LLM uncertainty estimation. The proposed method involves sampling multiple outputs from the model, measuring their pairwise similarity using another judge model, and averaging the similarity scores as the uncertainty measurement. The authors evaluate their method on a small synthetic dataset and the MATH benchmark, showing their method can predict the difficulty levels of data examples.

The reviewers recognize the potential applicability of the proposed method as it requires no access to internal states or parameters of the LLM (Reviewer Q3NW, h7Et, AGJq), clear writing (Reviewer 7xAU, h7Et, Bezx), and reasonable method design (Reviewer 7xAU, h7Et). However, all reviewers expressed their significant concerns. Specifically:
- Flawed and limited evaluation (reviewer Bezx, AGJq, h7Et, 7xAU, Q3NW). Reviewers are concerned regarding the questionable synthetic dataset and predict problem difficulty rather than model accuracy or calibration.
- Insufficient comparison to relevant prior work and baselines (Reviewers 7xAU, AGJq, Bezx). Reviewers also noted that TRUST is closely related to existing semantic uncertainty methods, making the claimed performance less convincing.
- Limited novelty and unclear technical contribution (Reviewers 7xAU, AGJq, Bezx). Reviewers also argued that the core idea of measuring semantic variance across temperature-sampled outputs is already well established.
- Weak and idealized theoretical analysis (Reviewers Q3NW, h7Et, AGJq, Bezx). The theoretical justification is limited to a highly simplified single-token setting, and reviewers think it is trivial or irrelevant to realistic scenarios.

**Reviewer Concerns:**

During the rebuttal phase, several reviewers actively engaged in further discussion on clarifying the intended definition of “uncertainty” and the relationship between TRUST, generation diversity, and prior semantic uncertainty methods. Despite the authors’ detailed responses and additional analyses, key reviewers (Bezx and AGJq) indicated that their core concerns regarding novelty, evaluation validity, and theoretical relevance remained unresolved and did not revise their scores. Reviewer h7Et acknowledged the additional experiments and clarifications but still emphasized disagreement with the authors’ stance that uncertainty can be treated as orthogonal to accuracy, maintaining a score below the acceptance threshold. Considering the consistently negative assessments across reviewers, this paper will benefit from substantial further improvements.

**Reviewer Scores:**

Reviewer h7Et, AGJq, and Bezx already fully engaged in the discussion and are unlikely to have significantly changed their scores. All of them have explicitly stated that their core concerns remained unresolved. For the remaining reviewers (Q3NW, 7xAU), given the strong overlap between their original concerns with the other three reviewers, it is unlikely that fuller participation would have led to a material increase in scores.

---

### Decision · Program_Chairs · 2026-01-26

Reject